# Illicit Drugs in Surface Waters: How to Get Fish off the Addictive Hook

**DOI:** 10.3390/ph17040537

**Published:** 2024-04-22

**Authors:** Halina Falfushynska, Piotr Rychter, Anastasiia Boshtova, Yuliia Faidiuk, Nadiia Kasianchuk, Piotr Rzymski

**Affiliations:** 1Faculty of Economics, Anhalt University of Applied Sciences, 06406 Bernburg, Germany; 2Faculty of Science & Technology, Jan Dlugosz University in Częstochowa, Armii Krajowej 13/15, 42200 Czestochowa, Poland; p.rychter@ujd.edu.pl; 3Nuffield Department of Medicine, University of Oxford, Oxford OX1 2JD, UK; a.boshtova@gmail.com; 4Hirszfeld Institute of Immunology and Experimental Therapy, Polish Academy of Sciences, Rudolfa Weigla 12, 53114 Wrocław, Poland; yuliia.faidiuk@hirszfeld.pl; 5Educational and Scientific Centre “Institute of Biology and Medicine”, Taras Shevchenko National University of Kyiv, 2 Prospekt Hlushkov, 03022 Kyiv, Ukraine; 6Zabolotny Institute of Microbiology and Virology, National Academy of Sciences of Ukraine, 154 Zabolotny Str., 03143 Kyiv, Ukraine; 7Faculty of Biology, Adam Mickiewicz University, 61712 Poznań, Poland; nadiia.kasianchuk@gmail.com; 8Department of Environmental Medicine, Poznan University of Medical Sciences, 60806 Poznań, Poland; rzymskipiotr@ump.edu.pl

**Keywords:** pharmaceutical pollution, toxic effects, fish physiology, environmental fate

## Abstract

The United Nations World Drug Report published in 2022 alarmed that the global market of illicit drugs is steadily expanding in space and scale. Substances of abuse are usually perceived in the light of threats to human health and public security, while the environmental aspects of their use and subsequent emissions usually remain less explored. However, as with other human activities, drug production, trade, and consumption of drugs may leave their environmental mark. Therefore, this paper aims to review the occurrence of illicit drugs in surface waters and their bioaccumulation and toxicity in fish. Illicit drugs of different groups, i.e., psychostimulants (methamphetamines/amphetamines, cocaine, and its metabolite benzoylecgonine) and depressants (opioids: morphine, heroin, methadone, fentanyl), can reach the aquatic environment through wastewater discharge as they are often not entirely removed during wastewater treatment processes, resulting in their subsequent circulation in nanomolar concentrations, potentially affecting aquatic biota, including fish. Exposure to such xenobiotics can induce oxidative stress and dysfunction to mitochondrial and lysosomal function, distort locomotion activity by regulating the dopaminergic and glutamatergic systems, increase the predation risk, instigate neurological disorders, disbalance neurotransmission, and produce histopathological alterations in the brain and liver tissues, similar to those described in mammals. Hence, this drugs-related multidimensional harm to fish should be thoroughly investigated in line with environmental protection policies before it is too late. At the same time, selected fish species (e.g., *Danio rerio*, zebrafish) can be employed as models to study toxic and binge-like effects of psychoactive, illicit compounds.

## 1. Introduction

In recent decades, a significant rise in the worldwide consumption of illicit drugs (IDs) has become evident. The “2019 EU Drug Markets Report” reveals that the annual retail sales of drugs exceeded 30 billion EUR, tightly related to the increased number of drug abusers. As shown, 50% of individuals aged 12 and older in the USA have used IDs at least once [1]. As of 2022, statistics indicate that 27% of adults aged 15–64 in EU countries have used cannabis at least once in their lifetime, while 5% and 4% have been exposed to cocaine and methamphetamines, respectively [2]. According to the World Drug Report, approximately 275 million individuals globally engaged in drug use in 2020. Additionally, over 36 million people have experienced drug use disorders. Moreover, an estimated 13.2 million individuals are identified as drug abusers, with 4.3 million inhabiting North America and 2.59 million located in Europe, accounting for 0.63% and 0.48% of the population correspondingly [3]. 

According to the US Centers for Disease Control and Prevention, since 1999, more than 1M fatalities have occurred in the United States due to drug overdose, with 106,699 drug overdose deaths recorded in 2021 alone. Of these, 75.4% were due to opioid overdose, although deaths involving psychostimulants such as methamphetamine were on the rise [4]. Considering the market value and continuous increase in ID usage, it is valuable to project a significant increase in detectable amounts of drugs and their active metabolites in waste and surface waters. Moreover, the COVID-19 pandemic has likely exacerbated the drug crisis, as highlighted not only by a higher number of substance use symptom records and an increased number of overdose fatalities but also by the elevated concentrations of selected psychoactive substances in wastewater, indicating increased drug use [5,6,7].

Environmental aspects usually remain in the shadow in the comprehensive exploration of IDs concerning their adverse impacts on human health and public security. However, as with any other human activity, drug production, trade, and consumption might imprint a significant footprint on the natural environment [8,9,10]. As individuals engage in drug consumption, legal or illegal, the residues are excreted into sewage systems through urine and feces. The proportions of consumed drugs that are excreted unaltered [11] and as metabolites in urine and feces are as follows: cocaine (COC; unaltered (1–9%) and its metabolite benzoylecgonine (BE; 35–54%); methamphetamine (METH; unaltered 43–62%); 3,4-methyl-enedioxy-methamphetamine (ecstasy, MDMA; unaltered 65%); and tetrahydrocannabinol (THC; unaltered 65–90% through feces and 10–25% through urine) [12]. Most of these drugs and their metabolites in raw wastewater can rarely be removed because the traditional wastewater treatment procedures are not specifically designed to reduce such compounds’ concentrations [13]. Therefore, the effluents from wastewater treatment plants (WWTPs) has become the major source of these drugs and their metabolites in the aquatic environment [14,15]. 

Although waste and surface waters include psychostimulants and depressants, a limited amount of research has focused on their eco-toxicological impacts on the aquatic environment. Being amenable to bioaccumulation in living organisms, IDs might accumulate in fish tissues even in low concentrations in the water column or sediments [13]. Drawing insights from data extrapolated from studies involving mammalian and human subjects, it becomes apparent that IDs possess the potential to induce a myriad of adverse effects, encompassing musculoskeletal and neurological disorders, as well as immune and endocrine disruptions, thereby making a substantial contribution to the decline in biodiversity [16]. It is therefore critical to look behind the scenes into the core chemical and biological processes and understand whether and how the use of these addictive compounds may affect non-target wildlife, as well as which negative outcomes are associated with them. With this purpose in mind, the present review provides a comprehensive analysis of the presence of IDs and their metabolites in water bodies. It also examines the accumulation of these substances in fish, particularly emphasizing the *Danio rerio (*Hamilton*,* 1822) species. Additionally, this review aims to explore the occurrence of IDs in the aquatic environment and the potentially hazardous effects of IDs on fish. It also discusses the promising biomarkers indicative of the toxic effects of these compounds.

## 2. Occurrence of Illicit Drugs in the Aquatic Environment

The excretion of IDs either in their unaltered form or as metabolites can lead to their entry into sewage systems and subsequent survival during WWTP processes. As a result, IDs can be found in various water bodies, including rivers, streams, surface water, and marine environments [17,18,19,20,21,22,23,24].

As WWTPs perform as the first barrier protecting mostly water ecosystems against released chemical compounds, various IDs, including amphetamine (AMP), MDMA, METH, COC, and BE, have been detected in influents of WWTPs worldwide, with COC being found in concentrations reaching up to 450 ng/L [25,26,27]. Furthermore, drugs with medical applications, such as codeine and morphine, have been observed at significantly higher levels, reaching approximately 6000 ng/L in influents (Table 1). While some studies have shown promising data regarding a decrease in active chemical loads of IDs in influents/effluents occurring over the years [20], many others have reported a general increase in consumption and subsequent entry into the environment, particularly of COC, METH, and ketamine (KET) [28]. Levels of AMP and MDMA mostly have fluctuated at similar levels or even slightly decreased when compared with recent years [29].

A distinct regional trend in drug concentrations is evident across Europe, particularly notable with BE, the primary metabolite of COC, detected at elevated levels in southern and western regions. In cities such as London and Bristol in the UK, Amsterdam in the Netherlands, Zurich and Geneva in Switzerland, and Antwerp in Belgium, concentrations ranged from 600 to 900 mg of BE/1000 people/day. Over a five-year investigation period, the concentration of this metabolite increased from 281–331 mg/1000 people/day in 2011–2013 to 329–373 mg/1000 people/day in 2014–2017. Conversely, AMP (approximately 40 mg/1000 people/day), MDMA (with a maximum of 33 mg/1000 people/day), and METH (declining from 39 mg/1000 people/day in 2011 to 18 mg/1000 people/day in 2013) were more prevalent in East and North–Central Europe [29,30]. Comparatively, METH exhibited the highest levels in the USA and Australia, while benzoylecgonine was most prominent in South America [30,31].

Efforts to remove IDs from WWTPs and decrease pharmaceutical concentrations in effluents have become a focus of research. It is reported that only a certain percentage of drugs are removed from municipal influents, and in countries where WWTP equipment is outdated, they hold dominant positions in the identification of IDs in surface water or maritime systems. Given the barrier role of WWTPs, wastewater treatment technologies must continually advance, employing sophisticated methods to effectively neutralize IDs and their metabolites [32]. Among the most promising methods are the reactor-activated sludge process [15], combined granular activated carbon and ozonation treatment, and anaerobic anoxic-oxic processes [33]. Unfortunately, there is a lack of publicly available data regarding the global implementation status of modern technologies in WWTPs for effectively removing pharmaceuticals from influents.
pharmaceuticals-17-00537-t001_Table 1Table 1Occurrence of selected illicit drugs in influents and effluents at water waste treatment plants in various countries.SubstanceCountryPlaceConcentration Range ng/LReferenceInfluentEffluentMedical-based illicit drugsMethadoneFranceCorbeil26.022.0[34]GermanyND130.0120.0[35]NetherlandsVarious cities<45.08.7–47.0[36]ItalyNosedo11.6 ± 1.79.1 ± 0.5[37]SlovakiaPetrzalka25 ± 318 ± 3.0[38]SPAINCatalonia3.4–1531.03.4–732[39]SwitzerlandZurich42.0–202.044.0–128.0[40]Logano49.7 ± 9.636.2 ± 2.8[37]United KingdomVarious locations171.168.8[41]England2.6–171.11.4–91.0[42]MorphineCosta RicaEl Roble67.0–77.015.0–61.0[43]GermanyND820.0110.0[44]440.029.0[35]MartiniqueFort de France154.0 ± 27.058.7 ± 7.0[45]SpainCatalonia25.5–278.012.0–81.1[39]Iberian Pensinsula54.2–166.05.4–80.5[46]Southeast region90.0–275.060.0–155.0[47]SwitzerlandLogano204.4 ± 49.955.4 ± 11.1[37]TaiwanTaipei38.029.0[48]USAVarious cities4523.0–1060.0<50.0[49]CodeineCosta RicaLiberia143.0–325.011.0–29.0[43]CyprusND2316.0–6460.0<LOD–3783.0[50]GermanyND540.0260.0[44]ItalyVerone275.0–335.0110.0–126.0[51]NetherlandsVarious cities240.0–536.0173.0–245.0[36]SlovakiaPetrzalka123.0 ± 11.024.0 ± 2.0[38]SpainSoutheast region234.0–1556.0289.0–786.0[47]United KingdomVarious locations2703.51206.2[42]TramadolGermanyND470.0370.0[35]SlovakiaPetrzalka860.0 ± 120.0710 ± 50[38]United KingdomND1320.7 ± 59.3506.0 ± 46.6[52]Illicit drugsCocaineAustraliaSouth-east Queensland20–200aND[53]ChinaBeijing32–562ND[54]BelgiumFlanders region22–687ND[55]CanadaEastern Canada (3 cities)209–823<LOQ–530[56]CroatiaVelika Gorica, Varazdin and Karlovac0.7–53ND[57]FranceParis21–1532ND[58]IrelandDublin’s surroundings48947–138[59]NetherlandsUtrecht, Eindhoven, Apeldoorn and Amsterdam87–957<6–159[36]AmphetamineChinaBeijing2–1010.9–2.8[60]BelgiumBrussels and Flanders region76aND[61]CanadaEastern Canada (3 cities)<LOQ–25<LOD–14[56]France Various cities125–194ND[58]NetherlandsUtrecht, Eindhoven, Apeldoorn and Amsterdam81–682<4–6.9[36]MethamphetamineChinaBeijing21.2–3040.4–12.2[60]BelgiumBrussels and Flanders region2ND[61]FRANCEVarious cities5124[58]ChinaHong Kong97.8ND[53]USANebraskaND350.1 ± 78.3[62]NetherlandsUtrecht, Eindhoven, Apeldoorn and Amsterdam<15–17<5[36]CanadaEastern Canada (3 cities)<LOQ–6517–95[56]HeroinBelgiumBrussels and Flanders region415ND[61]FRANCEVarious cities25–19431[58]LOD—limit of detection; LOQ—limit of quantification; ND—no data.


After undergoing partial removal in WWTPs or being directly discharged into water bodies through municipal effluents, IDs and their metabolites become prevalent in aquatic environments, emerging as significant pollutants of growing concern worldwide (Figure 1A,B). In France and the United Kingdom, rivers such as the Seine in Paris and the Thames have been found to contain notable concentrations of IDs, including METH, morphine, and codeine. Concentrations of these substances vary, with METH and morphine detected at less than 2 ng/L and codeine ranging from 3.1 to 13.8 ng/L in the Seine River [63], while in the UK, METH concentrations reached 38.2 ng/L, and morphine and codeine levels ranged from 0.5 to 42 ng/L and 341.7 to 5970 ng/L, respectively [41].

Figure 1A illustrates the widespread presence of COC, METH, and AMP in surface and tap water across continents. COC concentrations are variable, from European tap water samples (e.g., 18.6 ng/L in France to 166.7 ng/L in Portugal) to discharge points of wastewater treatment plants in Australia (e.g., 2990 ng/L at Redbank Creek) [65,66]. High MET concentrations are observed globally, with notable levels in the rivers of Taiwan and the USA (up to 25,250 ng/L in Hawkesbury-Nepean River) [66,67,68]. AMP is less frequently detected but still present, with traces found in European tap water and surface water in China and Taiwan [65,67,69].

Figure 1B illustrates the presence of methadone, morphine, codeine, and tramadol in the water of rivers of France, the United Kingdom, the Czech Republic, Germany, Spain, Italy, Switzerland, and China [44,70]. In France, the Seine River exhibited varying concentrations, with codeine ranging from 3.1 to 13.8 ng/L and tramadol from 12.4 to 90.3 ng/L [63]. UK rivers demonstrated methadone water concentrations up to 38.2 ng/L and significant levels of morphine and codeine. The Czech Republic’s rivers were shown to exhibit methadone and codeine water concentrations below 50 ng/L, contrasting with tramadol levels ranging from 38 to 663 ng/L [70]. Spanish rivers exhibited methadone and codeine below 31 ng/L, while Italian rivers displayed water concentrations under 40 ng/L [39,47]. Zurich’s waters contained methadone, morphine, and codeine water concentrations below 15 ng/L [40]. Chinese rivers showed methadone and codeine levels below 2 ng/L, while Taipei’s rivers exhibited concentrations up to 40 ng/L [48,71]. Notably, Lagos, Nigeria, displayed high tramadol concentrations, up to 852 ng/L [72].

Similarly, IDs have been found in rivers across several European countries, including the Czech Republic, Germany, Spain, Italy, and Switzerland (Figure 1A,B). In the Czech Republic, METH and codeine were detected in surface water at concentrations below 50 ng/L [70], while in Spain, morphine and codeine were found in various rivers, with concentrations ranging up to 174 ng/L for codeine [39,47]. Italian rivers such as the Olona, Lambro, and Po have also shown contamination with METH, morphine, and codeine, albeit at lower levels [73].

The presence of IDs is not limited to European waters; they have also been detected in rivers across Asia and Africa; however, the information is still limited (see Figure 1A,B). As an example, in Nigeria, a country facing significant environmental challenges due to population growth, urbanization, and inadequate waste management infrastructure, notably higher concentrations of pharmaceuticals, including IDs, have been observed in rivers [74]. The presence of these substances at µg/L underscores the severity of the contamination and the potential risks posed to aquatic ecosystems and human health. The high concentrations of pharmaceuticals in Nigerian rivers may be attributed to various factors, including inadequate wastewater treatment facilities, improper disposal of unused medications, and widespread pharmaceutical use without proper regulation or monitoring.

Although IDs have been found at low concentrations in surface waters, their continuous introduction into the marine environment emphasizes their persistence and potential adverse effects. While the concentration of AMP and METH in water samples remains relatively low, not exceeding 40 ng/L, COC has been found at significantly higher levels, reaching up to 537 ng/L (Figure 1A, Table 2). Despite their medium Log Kow (octanol–water partition coefficient) of around 2.0, indicating a low-affinity accumulation within cells [75], there remains a possibility of bioaccumulation and adverse effects on aquatic biota, especially considering the higher vulnerability of marine biota compared to freshwater counterparts.

In summary, despite being detected at low concentrations in surface waters, the continuous intrusion of IDs into the marine environment underscores their persistence and potential adverse effects. Medical-prescribed codeine and morphine, as well as drugs of abuse, e.g., COC, have been found at significantly higher levels compared to other IDs, raising concerns about their potential bioaccumulation and its impact on aquatic biota.

## 3. Bioaccumulation of Illicit Drugs and Their Toxicity

### 3.1. Bioaccumulation of Illicit Drugs in Fish

The accumulation of IDs and their metabolites in aquatic ecosystems has garnered significant attention due to its potential impact on wildlife and, by extension, human health. Furthermore, IDs can also accumulate in sediments, posing risks to sediment-dwelling organisms and potentially leading to long-term exposure issues [81]. This adds a critical dimension to our understanding of environmental contamination, emphasizing the complexity of pharmaceutical pollution beyond water bodies alone. Numerous studies across various species have documented the persistence of these substances in fish tissues, especially in the brain, kidneys, muscles, and liver [82,83,84,85,86,87]. These organs serve as critical matrices for assessing the ecological risks pharmaceuticals pose to aquatic environments [84,85,86,88]. The relationship between the concentration of drugs in water and their accumulation within fish highlights the direct impact of environmental exposure on aquatic life.

Building upon this foundation, the bioaccumulation of drugs of abuse threatens aquatic organisms with chronic or acute harmful effects through processes such as assimilation, translocation, and biomagnification in nutrient cycles. It also extends its impact to apex predators, including humans who consume fish. This accumulation chain underscores the urgency for further research into the ecological risks and consequences arising from ID residues in water systems, as noted by [13,23] and others. The interconnectedness between the bioaccumulation in fish tissues and the broader ecological implications necessitates a comprehensive investigation to mitigate potential risks to aquatic life and, by extension, human health.

In the study conducted by Yin et al. [84], zebrafish were subjected to a semi-static exposure system with water renewed every 24 h, where they were exposed to a mixture of METH and KET at three different concentrations (0.01 μg/L, 1.00 μg/L, and 100.00 μg/L) for 14 days, followed by a 7-day depuration period in clean water. This approach revealed a consistent presence of KET in zebrafish tissues, with the concentrations of KET during the uptake phase being 0.011 ± 0.0019 μg/L, 0.9 ± 0.1 μg/L, and 96.1 ± 16.7 μg/L for the low, medium, and high exposure groups, respectively. Notably, in the high treatment group, KET concentrations reached up to 986 ng/g wet weight in the brain, underscoring the drug’s significant bioaccumulation potential. The order of mean concentrations across various tissues was observed as brain ≈ liver > intestine > ovary > muscle, indicating a differential accumulation pattern that mirrors the external drug concentrations. The concentration of the KET derivative, norKET, was significantly lower than that of KET itself, likely due to limited biotransformation of KET to norKET or a lower bioaccumulation potential of norKET. Furthermore, the same study found METH concentrations in zebrafish tissues to follow a similar pattern of distribution, with a higher potential for accumulation in the brain, where the mean concentration of METH in fish exposed to the highest dose (100 μg/L) reached up to 425 ng/g wet weight. The concentration of the METH derivative AMP, which was consistently detected in fish tissues, was also highest in the brain of the low and intermediate-exposure groups. However, in the high-exposure group, the mean concentration of AMP in the ovary (553 ng/g) was almost twice that in the brain, and the concentrations in the liver, intestine, ovary, and muscle were even higher than those of METH. The ovarian AMP accumulation potential increased with increasing exposure to METH concentrations. All in all, these results may indicate that AMP is more ecotoxic than its parent compound METH, with the ovaries being a target organ of AMP, which may potentially harm fish reproduction.

Another piece of research further elucidated that METH could be remarkably accumulated in fish brains, with the distribution factor versus water standing at an impressive 232.5-fold. Notably, METH exposure was found to cause physiological function disorders in the fish, affecting aspects such as swimming trajectories, locomotion distances, and feeding rates. Additionally, METH stimulated surfacing behavior in loach, indicating a significant impact on the behavioral patterns of the fish [89].

Both KET and METH were rapidly uptaken by the zebrafish, with uptake rate constants ranging from 0.6 to 1.4 × 10^3^ L/(kg/d), and eliminated with elimination rate constants ranging from 0.2 to 7.0 1/d, resulting in short half-lives of 0.2 to 7.0 d. While *Ku* values of both substances decreased with the increasing exposure concentrations, *Ke* values increased, which may be indicative of metabolic changes in zebrafish. The bioconcentration factors (BCFs) for both KET and METH in fish tissues exhibited a decrease with increasing exposure concentration, suggesting mechanisms like saturation of uptake or elimination processes might be at play. Specifically, BCF values for KET and METH showed a decreasing trend (78.7 ± 0.7 > 5.4 ± 0.2 > 3.0 ± 0.1 and 122.0 ± 5.6 > 3.0 ± 0.1 >1.1 ± 0.1 correspondingly) across low, medium, and high exposure groups, respectively, a phenomenon that could be attributed to saturation of uptake/elimination processes, functional impairment of metabolism, and the excretion and biodegradation of toxicants from the fish body [84,90]. 

Building on this evidence, extensive research further corroborates the tendency of KET to bioaccumulate in aquatic species. For example, a study by Wang et al. (2020) [83] on adult medaka *Oryzias latipes* (Temminck & Schlegel) demonstrated significant bioconcentration of KET, with factors ranging from approximately 1.07 to 10.94 after 90 days of exposure to both environmentally relevant (0.05–0.5 μg·L^−1^) and higher concentrations (5–100 μg·L^−1^). This resulted in body concentrations of KET ranging from 0.55 ± 0.07 to 113.00 ± 2.12 ng/g wet weight, directly correlating with the exposure levels. NorKET, a derivative of KET, was detected only at higher exposure levels, with body concentrations of 2.00 ± 0.35 to 54.00 ± 0.71 ng/g wet weight, indicating a selective bioaccumulation process. This study also observed histological abnormalities and behavioral function abnormalities in the fish, suggesting potential adverse effects of such bioaccumulation.

COC and its metabolites have also been found to accumulate in fish, with studies demonstrating that European eels *Anguilla anguilla* L. can accumulate COC, leading to alterations in various organs and systems [19,85,91,92,93,94]. Capaldo et al. [85] documented the accumulation of COC in European eels after exposure to environmentally relevant COC concentrations of 20 ng/L for one month, finding the highest concentrations in the brain (30.5  ±  0.4 pg/g), muscle (20.2  ±  0.5 pg/g), liver (13.4  ±  2.2 pg/g), and kidney (11.4  ±  1.2 pg/g). This accumulation suggests a high affinity of nervous tissue for COC and points to the fat content in these tissues as a potential factor for higher COC levels [85,95]. The high levels of COC found in kidney tissue may be attributed to the involvement of the kidneys in COC metabolism [85].

Kirla Krishna Tulasi et al. (2016, 2017) [87,96] reported a significant accumulation of COC (1516 ± 84 mg/kg) in zebrafish larvae following an 8 h exposure, a level that notably persisted even after 48 h of depuration (1331 ± 98 mg/kg). This enduring presence is likely attributed to the high affinity of COC for melanin, highlighting the substance’s tenacity within biological systems. The study also observed high COC concentrations in the brain (406 ± 44 mg/kg) and the trunk (255 ± 32 mg/kg). However, those dropped to very low concentrations following the depuration phase (brain (50 ± 3.7 mg/kg) and trunk (44 ± 2.2 mg/kg). Norcocaine, the biotransformation product of COC, was detected in the analyzed body parts after 8 h of COC exposure at a concentration of approximately 0.6% compared to that of COC. In the whole larvae, the concentration varied from 0.5% to 3.6% at different time points, following a distribution pattern similar to that of COC [87]. In a complementary study, Ondarza et al. [86] conducted an assessment of wild fish samples from two distinct locations: the Acaraguá and Paraná rivers in Argentina. This investigation identified benzoylecgonine, the primary COC metabolite, in the muscles of thararira (0.87 μg/kg ww) and the gills of catfish (1.60 μg/kg wet weight). Furthermore, traces of BE were detected in the liver (0.5 ± 0.7 μg/kg wet weight) and gills (0.4 ± 0.6 μg/kg wet weight) of female streaked prochilod *Prochilodus lineatus* (Valenciennes, 1837) from the Acaragua and Parana rivers, respectively.

Additionally, the interplay between IDs and other environmental contaminants, such as microplastics, further complicates their ecological impact. Research by Qu et al. (2020, 2022) [97,98] has illustrated that the co-presence of METH and MPs not only exacerbates the acute toxicity experienced by algae and snails but also hampers algal cell proliferation. This interaction between IDs and microplastics highlights a synergistic effect that can amplify the adverse outcomes on aquatic life. Moreover, microplastics have been found to influence IDs’ enantioselectivity, altering their toxicological effects and environmental fate.

These studies collectively underscore the critical need for comprehensive research to understand the full scope of ecological risks posed by the presence of IDs residues in water systems. The evidence of drug accumulation in fish tissues and observed physiological and behavioral changes call for an urgent assessment of potential environmental and health impacts. The interaction between IDs and other environmental contaminants, such as microplastics, further complicates this issue, highlighting the need for comprehensive studies to understand the multifaceted impacts of these substances on aquatic ecosystems and human health.

### 3.2. Behavior Disorders in Fish as the Frontline Response to Illicit Drugs

Exploring behavior disorders in fish as an initial reaction to IDs is a vital field of research in aquatic toxicology, as such alterations can have profound and far-reaching implications for aquatic ecosystems. That is to say, swimming performance is a crucial factor in predator–prey interactions, reproductive success, food capture, growth, and, ultimately, fish survival in their natural environment. When external substances disrupt these behaviors, they affect individual survival and can lead to significant shifts in the composition and functionality of aquatic communities over time. The significance of these studies lies in their potential to reveal the evolutionary and environmental consequences of drug pollution, informing conservation strategies and policy decisions aimed at preserving the delicate balance of aquatic ecosystems [99,100].

METH exposure has been linked to significant behavioral alterations in fish. For instance, Sancho Santos et al. (2023) observed that brown trout (*Salmo trutta* m. *fario* L.) exposed to environmentally relevant concentrations of METH (1 μg/L) for 28 days displayed reduced activity levels [101]. Similarly, Liao et al. (2015) reported dose-dependent effects of METH on zebrafish, inducing hypolocomotion and affecting swimming metrics such as active time and average velocity. Intriguingly, while low doses of METH-induced a startle response, this effect was alleviated at higher concentrations (10–40 μM), suggesting complex dose-dependent behavioral responses [102]. Moreover, METH exposure led to a peculiar clockwise movement tendency in zebrafish larvae under prolonged exposure.

In their study, Horký et al. (2021) reported on the dependency behaviors observed in brown trout subjected to METH at concentrations deemed environmentally relevant (1 µg/L) [103]. These fish displayed a marked preference for METH, a behavior that continued for up to four days post-cleansing period. The presence of AMP was confirmed solely in the brain tissues of these trout, showing a decrease in detection from all individuals to just 12.5% throughout the 10-day period following exposure. This decrease was directly linked to the fishes’ predilection for METH, as evidenced by the correlation between METH preference and the residue levels of AMP in their brains. Additionally, these trout exhibited a diminished likelihood of movement during the withdrawal phase, a trend that persisted for four days and closely paralleled the concentrations of AMP in their brains. Conversely, an uptick in movement was noted upon the identification of METH in the brain tissues, indicative of increased activity levels. The frequent observation of specific individuals in the area of the observation arena where dosing occurred implies that the detected presence of METH in the brain stems from immediate METH consumption.

In examining the impact of KET on aquatic organisms, research has consistently demonstrated that exposure to even low concentrations within the range of environmental relevance can significantly influence the behavior and physiology of zebrafish and medaka. Specifically, alterations in swimming behaviors, including changes in total swimming distance, velocity, and feeding rates, have been documented. Notably, Wang et al. (2020) observed that adult medaka fish showed a marked increase in feeding rates at KET concentrations starting from 0.05 μg/L, along with disruptions in food preferences at concentrations of 0.5 μg/L and above [83]. This behavior suggests increased metabolic activity and energy demands among the exposed fish groups.

Further investigations have identified a stimulatory effect of low concentrations of KET (0.05–100 μg/L) on both zebrafish and medaka, with notable increases in movement distance and speed following exposure to sub-anesthetic doses ranging between 2 and 40 mg/L [89,104,105]. Intriguingly, these same sub-anesthetic doses elicited a reduction in anxiety levels among zebrafish larvae, indicating a nuanced dose–response relationship that varies between adult and larval stages of development [89,104].

Other studies, including those by Zakhary et al. (2011) and Kanungo et al. (2013), have highlighted the dose-dependent nature of KET’s effects on behavior and motor neuron toxicity in both wild-type and transgenic zebrafish models. These effects are thought to stem from KET’s impact on differentiating neurons. Interestingly, KET-induced alterations in acetylcholinesterase expression or neurotoxicity might disrupt larval locomotion—a phenomenon similarly observed when the neurotoxic pesticide endosulfan inhibited brain acetylcholinesterase activity, leading to impaired swimming performance in adult zebrafish [106]. Furthermore, Felix et al. (2017) reported that KET exposure (200–800 mg/L, 20 min) during critical early embryonic stages—specifically at 50% epiboly and 1–4 somites—leads to a significant uptick in anxiety-like behaviors and a reduction in avoidant behaviors, with embryos also showing a dose-dependent decrease in average speed. This underscores the sensitivity of developmental stages to KET exposure and its potential to impact early life behaviors.

COC exposure in zebrafish presented a different pattern. While non-anesthetic doses of the substance (0.015–15 μM) did not immediately alter locomotor activity, the period following COC withdrawal exhibited a notable increase in behavioral hyperactivity. This heightened activity began to escalate during the initial 24–72 h of withdrawal and persisted for at least five days. Such observations led to the suggestion that COC withdrawal induces a prolonged anxiety-like effect in zebrafish, highlighting the complex behavioral impacts of this drug on aquatic organisms [107].

These studies provide a comprehensive picture of IDs’ complex and multifaceted effects on aquatic life. From altering fundamental behaviors such as swimming and feeding to inducing addiction-like states and affecting developmental stages, IDs pose significant challenges to the health and stability of aquatic ecosystems.

### 3.3. Illicit Drugs-Induced Oxidative Stress and Metabolic Changes in Fish

#### 3.3.1. Metabolic Alterations in Fish Exposed to Illicit Drugs

Numerous investigations reveal that even at an environmental level, METH has significant impacts on fish in terms of their metabolism. As an example, brown trout (*Salmo trutta* m. *fario*) that were exposed to 1 µg/L METH for 28 days showed reduced levels of metabolic rates. These changes were shown to be associated with the concentrations of METH and AMPH in the brain, as well as changes in brain metabolomics [101].

Evidence has demonstrated that METH increases melatonin levels and activates associated pathways in the brains of brown trout [101]. This phenomenon may be attributed to melatonin’s antioxidant properties, including direct free radical scavenging and increasing the efficiency of mitochondrial oxidative phosphorylation and reducing electron leakage and its capability to diminish the proinflammatory mediators, which serve to protect the nervous system and mitigate the detrimental effects of METH [108,109]. The therapeutic potential of pre-treating with melatonin to mitigate adverse outcomes associated with METH, such as METH-induced pathological changes resembling Alzheimer’s disease and dysfunction of the blood–brain barrier, was also emphasized for murine models [110,111]. 

Furthermore, METH did not significantly alter the levels of monoamine neurotransmitters in the zebrafish brain but did lead to reductions in L-tyrosine and L-carnitine [112]. L-tyrosine serves as a precursor for neurotransmitters such as dopamine, norepinephrine, epinephrine, and octopamine and is recognized as a key biomarker for amino acid metabolism, suggesting potential disruptions in this metabolic pathway within the fish brain. The disturbance observed in L-carnitine levels suggests that the fish brain experienced disruptions in energy metabolism, particularly concerning the transfer of long-chain fatty acids across the inner mitochondrial membrane and fatty acids degradation via carnitine palmitoyltransferase I [113]. Similar findings regarding energy deficit were observed in the mouse striatum, potentially related to metabolic stress induced by METH on dopaminergic neurons, attributed to the release of dopamine and direct inhibition of the mitochondrial respiratory chain [114].

METH exposure leads to significant changes in lipid composition across brain regions in both zebrafish larvae and mammals, as evidenced by studies [115,116]. However, investigations into the mechanistic underpinnings of lipid metabolism alterations in fish remain largely overlooked. Specifically, there is a need for in-depth exploration of sphingomyelins and monoglycosylceramides, which are abundant in presynaptic membranes, play a crucial role in regulating the timing of synaptic vesicle fusion, and are affected by heavy METH abuse in human subjects [117]. Furthermore, these lipid profile alterations can potentially disrupt synaptic transmission by influencing ion channels, signaling pathways, and neurotransmitter release. Consequently, such disruptions may contribute to the manifestation of behavioral abnormalities, a phenomenon that warrants further exploration in fish models.

As seen in rodents, METH after intraperitoneal injection of 40 mg/kg increases dopamine and glutamate levels in the zebrafish brain but decreases 5-hydroxytryptamine levels, which may be associated with the initiation of the destruction of 5-hydroxytryptamine (serotonin) nerve terminals [118]. An increase in dopamine and norepinephrine (but with lower variability) and a decrease in serotonin was also observed in the brains of rosy bitterling *Rhodeus ocellatus* (Kner, 1866), loach *Paramisgurnus dabryanus* (Dabry de Thiersan 1872), and mosquito fish *Gambusia affinis* (Baird & Girard, 1853), and particularly intense in Chinese medaka *Oryzias sinensis* Chen, Uwa & Chu, 1989 fish (up to 2.5-fold) after 20 days of waterborne METH exposure (25 µg/L) [89]. It should be noted that METH-induced lethality in zebrafish larvae in a dose-dependent manner in the range of concentration from 500 μM to 50 mM for 5 h, which was linked to initial sympathomimetic activation mediated by the dopaminergic system [119]. Despite being less sensitive to the effects of METH compared to serotonergic systems, the dopaminergic system’s activation could potentially have detrimental effects on the health status of fish and even lead to biodiversity decline.

Following METH exposure, zebrafish showed a biphasic response characterized by dysregulated cAMP and Ca^2+^ levels. This response manifested as sustained reductions in heart rate, initial increases in heart rate variability followed by a decline in QTc intervals, and notable fibrotic responses [120].

While research on the effects of METH on fish is somewhat explored, others, among them COC, have largely been overlooked. Notably, a study found that exposure to 20 ng/L of COC for 50 days increased blood glucose concentration in eel *Anguilla anguilla*, which remained unchanged even after a 10-day post-exposure recovery period [121]. This reaction could indicate an increased energy demand needed to cope with chemical-induced stress. It is noteworthy that METH, conversely, inhibits glucose uptake in abusers and induces hypoglycemia [122]. COC, on the other hand, is recognized for its ability to raise plasma glucose levels by stimulating the release of catecholamines [123]. However, it is unclear whether a similar mechanism occurs in fish because, as demonstrated in zebrafish larvae, COC decreased dopamine plasma concentration [124]. Thus, further in-depth studies are warranted.

Exposure to subacute concentrations of COC (0.3 μg/L and 1.0 μg/L) and its primary metabolites, benzoylecgonine and ecgonine methyl ester, induced shifts in the protein composition of zebrafish embryos at 96 h post-fertilization [125]. These alterations impacted the levels of proteins involved in lipid transport, lipid and energy metabolism, cytoskeletal organization, and response to oxidative stress. Analysis of the significant changes in protein levels compared to the control suggests that COC metabolites may exhibit higher toxicity than COC itself, irrespective of the concentrations examined [125].

#### 3.3.2. Oxidative Stress as the Marker of Illicit Drugs in Fish

There is a growing understanding that oxidative stress could be a contributing factor to the toxicity associated with METH and COC in higher vertebrates and humans [126,127]. As mentioned in Section 3.3.1, drugs of abuse stimulate the release of dopamine, which is quickly oxidized spontaneously or metabolized by enzymes like monoamine oxidase B and COX1/2 [127]. This metabolic process primarily occurs in the cerebral capillaries, producing superoxide ions and hydrogen peroxide, thus elevating oxidative stress levels [126,127].

Considering that the central nervous system in vertebrates, including fish, exhibits high oxygen consumption and relatively limited antioxidant defenses [128], it becomes evident that any disruption in the balance of reactive oxygen species could have significant consequences. Given that the central nervous system is enriched with cholesterol and sphingolipids, it becomes exceptionally vulnerable to oxidative stress. Therefore, it is plausible to expect that IDs in fish could induce considerable oxidative stress. However, the information regarding the antioxidant–pro-oxidant balance in fish following METH and especially COC exposure remains limited.

Acute exposure to METH at a concentration of 500 μg/L, either alone or in combination with microplastics, led to a significant decrease in the expression of antioxidant enzyme genes in zebrafish larvae [129]. Notably, zebrafish larvae exhibited a high sensitivity to these exposures, with a clear correlation observed between the activity of antioxidant enzymes and the mortality rate [129]. This susceptibility could potentially be attributed to the disruption of mitochondrial energy metabolism, similar to what has been observed in mammals [130]. Specifically, both the Krebs cycle and the electron transport chain may be blocked, leading to alterations in mitochondrial biogenesis, mitophagy, and fusion/fission processes [130]. These alterations have been shown to contribute to dopaminergic toxicity induced by METH. METH-induced impairment of mitochondrial function might also increase susceptibility to oxidative stress, promote programmed cell death, and exacerbate neuroinflammation [130]. Our previous research has demonstrated that zebrafish mitochondria are susceptible to various stressors, and the concept of “metabolic arrest” plays a pivotal role in their survival when exposed to a wide range of organic pollutants, including pharmaceuticals and cyanotoxins [131,132,133,134,135]. Therefore, it is plausible that similar mechanisms observed in mammals may also occur in zebrafish, although experimental verification is needed.

Contrastingly, adult zebrafish exposed to METH concentrations ranging from 20 to 40 mg L^−1^ for 48 h did not exhibit signs of lipid peroxidation in the brain. However, there was a noticeable down-regulation in tyrosine hydroxylase 2 expression, along with a concentration- and time-dependent decline in the levels of dopamine, norepinephrine (NE), and serotonin [136].

Interestingly, lower concentrations of METH (ranging from 0.05 to 100 μg/L) had a different effect, acting as inducers of antioxidants in the fish brain. Specifically, it was found that METH concentrations in the range of 0.05 to 100 μg/L, when administered for a period of 90 days, triggered the activation of the mitogen-activated protein kinase signaling pathway associated with extracellular signaling kinase 1 and protein kinase A in Japanese medaka. This activation was observed alongside minimal signs of gliosis, neuronal loss, and necrosis in brain tissues [137]. 

Based on our knowledge, only one study has demonstrated that even at environmental concentrations (ranging from 0.04 to 40 nM), COC and its metabolites, benzoylecgonine, and ecgonine methyl ester, induce the activation of superoxide dismutase and glutathione peroxidase in zebrafish (*Danio rerio*) embryos at 96 h post-fertilization. This activation is accompanied by an increase in reactive oxygen species production and genetic damage, as evidenced by DNA fragmentation and the appearance of micronuclei. However, the observed changes in enzymatic activity and expression of corresponding antioxidants did not show consistent agreement [125,138].

So, the toxicity of METH and COC in fish may be linked to oxidative stress. These drugs stimulate dopamine release, which might produce reactive oxygen species, among them superoxide ions and hydrogen peroxide, leading to lipid peroxidation and DNA damage. Nevertheless, it is important to acknowledge that our understanding of the antioxidant–pro-oxidant equilibrium in fish after exposure to METH and COC is still limited. Therefore, further research is essential to fully understand the impact of these drugs on oxidative stress and antioxidant defense mechanisms in aquatic organisms.

### 3.4. Endocrine-, Immune-Modulatory, and Inflammatory Effects of Illicit Drugs in Fish

Recent findings indicate that COC exposure can influence the hypothalamic–pituitary–adrenal axis in various species, including humans, non-human primates, rodents, and eels [139]. COC is known to increase plasma adrenocorticotropic hormone and cortisol/corticosterone levels, implicating oxytocin, arginine–vasopressin, and corticotropin-releasing hormone in this process. Chronic COC exposure in eels resulted in decreased adrenocorticotropic hormone and corticosterone levels but elevated cortisol levels, indicating potential dysregulation in the hypothalamic–pituitary–adrenal axis [139,140]. This dysregulation could have significant consequences, as cortisol elevation is often associated with stress in fish, which may disrupt reproductive processes such as migration and gonadal maturation [140]. Additionally, exposure to METH at environmentally relevant (0.05–5 μg/L) and high concentrations (25 and 100 μg/L) for 90 days has been linked to developmental abnormalities in fish offspring and high risk quotients values (>1), highlighting concerns about transgenerational toxicity [137]. High concentrations of METH in surface water have raised teratogenic risks, indicating potential adverse effects on wild fish populations. Furthermore, synthetic hallucinogens such as the NBOMe family compounds, including 25C-NBOMe (5-HT2A receptor agonists), have been implicated in developmental abnormalities in fish embryos, including alterations in trunk length, body curvature, and yolk formation [141]. These effects align with known teratogenicity indicators associated with exposure to various substances, including changes in muscle structures and cardiovascular abnormalities [120].

Adverse effects of METH on the central nervous system of zebrafish indicate its potential to impact the endocrine system in these aquatic organisms, similar to other studied compounds [112]. The impact of these disruptions can be far-reaching, affecting various aspects of zebrafish biology, from behavior and metabolism to reproductive and developmental processes [84,112,134,142,143]. The endocrine regulatory effect of METH exacerbates through its impact on cognitive functioning, neurotransmitter systems, and metabolic pathways [144]. The specific endocrine-disrupting mechanisms of IDs such as METH, COC, and its primary metabolites benzoylecgonine and ecgonine methyl ester on fish, nevertheless, remain elusive. In the case of COC exposure, alterations in vitellogenin patterns may elucidate a potential impact on the endocrine system. COC exposure in zebrafish induced significant changes in vitellogenin, particularly type I vitellogenin [125], which plays a vital role in supplying nutrients and energy during embryo development. Vitellogenin undergoes proteolytic cleavage, producing two lipovitellins and one phosvitin [145], essential for ensuring adequate nutrient and energy supply and immunity throughout embryo development [146,147]. Since changes in vitellogenin patterns are commonly associated with substances that disrupt the endocrine system [132], further research is needed to uncover the precise mechanisms underlying the endocrine-modulatory effects of IDs on fish and their implications for aquatic ecosystems. Additionally, there is evidence of widespread endocrine disruption in fish, with estrogenic effects demonstrated in various aquatic environments [148]. Furthermore, the role of cytochrome P450 enzymes in the catalyzation of COC N-demethylation in fish liver has been investigated, shedding light on the metabolic pathways of COC in fish [149]. Additionally, lead and other trace elements have been shown to negatively affect endocrine function in fish, further emphasizing the potential for environmental contaminants to modulate the endocrine system in aquatic species [150].

In terms of the immune system, there remains a lack of information regarding the effects of IDs on fish despite the significant sensitivity of immune-related parameters to both conventional and emerging pollutants [75,131,133,134,135]. Mentioned earlier alterations in fish brain metabolomes by environmental concentrations of METH, namely such specific metabolites as lyso-phosphatidylethanolamine, lyso-phosphatidylcholine, amino acids, carnitine, and long-chain fatty acids may be related to inflammatory responses [103,112,129]. As METH is a substrate of the dopamine transporter and taking into consideration the interconnectedness of the immune and nervous systems, it is plausible that IDs may exert immunomodulatory effects in fish through their impact on neurotransmitter systems and physiological processes [151]. Moreover, the presence of immunomodulatory substances in fish, such as antimicrobial peptides, additionally indicates the existence of a neuro-immune interaction in fish, which could be influenced by external factors like METH or COC exposure [152,153]. It has been demonstrated that METH exerts detrimental impacts on the immune systems of mammals and humans, causing substantial changes in both pro-inflammatory and anti-inflammatory cytokines. METH immunomodulatory effects can be exacerbated via either immunosuppression, including alterations in immune cell function and decreased immune response, inhibition of T cell proliferation, alterations in antibody production, decrease in CD4+ T cell frequency [154,155,156], or inflammation. METH-induced cytokine production promotes B cell infiltration into tissues and reduces systemic antibody levels following its accumulation in cells, consequently triggering a series of signaling reactions in both innate and adaptive immune cells [157,158]. Pro-inflammatory cytokine production, including interleukin-6 and tumor necrosis factor-alpha, may lead to a neuroinflammatory response, which may be directly connected with neurotoxicity and brain tissue damage (neurodegeneration) due to pro-inflammatory activation of brain innate immune cells, microglia [89,159], or induce inflammation in other tissues. Hand-in-hand with the immune response, inflammation emerges as a significant consequence, affecting both innate and adaptive immune functions in fish. This disruption can create an inflammatory environment within the central nervous system, impacting glial cells and perpetuating immune disorders. METH-induced neuroinflammation triggers long-term gene expression changes in brain regions associated with reward processing and drug-seeking behavior and, therefore, may be responsible for behavioral responses to the drug [160]. Additionally, the combined exposure to microplastics and METH exacerbates inflammatory responses in zebrafish larvae, leading to oxidative stress and alterations in inflammation-related gene expression [116]. Reported significant alterations in other cytokines, interleukin-10, interleukin-1β, and cyclooxygenase-2, are crucial in both resisting infections and the progression of METH-induced neurotoxicity [157,158]. 

The activation of the NLRP3 inflammasome and Toll-like receptors (TLRs), linked to infection and inflammation, may also play pivotal roles in METH addiction [158]. NOD- and TLR-signalling pathways might be involved in the response of fish to the adverse effects of drug exposure [161], although direct translation of data from mammal systems to fish should be performed with caution. Even though the *Danio rerio* model was utilized for studying hematopoiesis and drug transporters, making it a valuable tool for shedding light on inflammatory and toxicological responses [162,163], and it is considered as a relevant model for inflammation due to its high degree of similarity to mammal genome, there still remains a gap. Fish may have incomplete or alternative cell signaling networks due to the absence of some homologs of mammal signaling receptors and other cascade proteins, and some might be under-characterized due to poor annotations of existing genomes [129,164]. Therefore, the recognition mechanisms for IDs and downstream signaling cascades may differ. For instance, a homolog of TLR4, a major receptor responsible for METH recognition, is absent in a large number of fish species. Its function can be exerted via alternative signaling cascades; however, it is less efficient. For instance, as it is also a receptor for bacterial LPS (endotoxin), the recognition of this molecule in fish leads to inflammation only if endotoxin is present in much larger concentrations (µg/mL) than in the case of mammals (ng/mL). Additionally, the accessory molecules of the pathway, like Myeloid differentiation protein 2 (MD2), have not been identified in fish [165] until a recent report by Loes et al. [166] on the identification of the MD-2 coding gene *ly96* in a small fraction of macrophage-like cells using single-cell RNA seq. 

Another substance that is also supposed to be recognized via the TLR4 receptor is cocaine. Similarly to METH, it has a proinflammatory effect, which is rather systematic [167,168]. Neurodegeneration, as a result of neuroinflammation, is observed, together with the toxicity effect exerted on neurons. The immunomodulatory effects of cocaine in fish have been a subject of interest due to its potential impact on the innate immune system. While limited studies have been conducted on this topic, it is known that cocaine can induce various physiological changes in fish, such as altering heart rate and electrocardiogram parameters [169]. COC exposure has been shown to affect the dopaminergic system and anxiety levels in zebrafish, indicating its potential to modulate neurotransmitter function [170,171]. Cocaine is responsible for the blockage of the dopamine transporter, increasing the levels of dopamine in the synaptic cleft [127]. Serotonin (5-HT) and 5-HT receptors’ modulatory role in the mechanisms of action of COC were studied in serotonin transporter knockout rats [168]. The entry of COC through the fish skin and gills has resulted in an anesthetic effect, overriding potential stimulatory responses in the brain [87]. These findings suggest that COC may have complex effects on fishes’ neuroendocrine and cardiovascular systems, which could potentially impact their immune responses.

COC has been associated with the modulation of tight junction protein expression and the CCL2/CCR2 axis across the blood–brain barrier, the signaling axis involved in macrophage recruitment. A CCR2 orthologue was identified in zebrafish, supported by functional evidence [172]; therefore, its involvement in cocaine-induced cell signaling in fish cannot be excluded. Studies have demonstrated that COC exposure can lead to the downregulation of miR-124, a molecule reported to be produced by microglial cells and promoting neuroinflammation [173]. This molecule is also detected in fish, although its specific role in fish is still to be determined [174].

The longitudinal effects of embryonic exposure to COC in zebrafish have been explored, revealing a bell-shaped dose–response curve, suggesting that pre-exposure may make fish more sensitive to the drug as adults [175]. Moreover, COC exposure in eels notably boosts liver and kidney cyclooxygenase activity, signaling immune system activation [121]. The kappa-opioid receptor/dynorphin system has been found to have counter-modulatory effects on the reward caused by COC exposure, indicating a complex interplay of neurobiological systems in response to COC [176]. 

All those findings highlight the significant impact of IDs on fish immune and endocrine systems, underscoring the necessity for in-depth studies into mechanisms that combat emerging pharmaceutical pollution, particularly drugs of abuse. Comprehensive strategies are important to mitigate aquatic environmental contamination and protect vulnerable ecosystems.

### 3.5. Geno- and Cytotoxic Effects of Illicit Drugs in Fish

Although IDs’ direct teratogenic, genotoxic, and cytotoxic effects on human health are well-documented, emerging evidence suggests that these substances may also present significant risks to aquatic ecosystems, particularly fish populations. Research indicates that drug abusers exhibit elevated frequencies of nuclear abnormalities, including nuclear buds, binucleated cells, pyknotic nuclei, karyorrhexis, and abnormally condensed chromatin when compared to healthy controls [177]. These abnormalities may be associated with reactive oxygen species, which are produced by cytochrome P450, including P4503A and P4502B isozymes, during the metabolism of IDs. Additionally, altered DNA-methylation and DNA-hydroxy-methylation mechanisms have been observed, along with deacetylation-related epigenetic effects [178]. 

While still limited, recent research has emphasized various IDs’ genotoxic and cytotoxic effects on fish, underscoring the need for increasing awareness and environmental protection measures. One area of concern is the impact of METH exposure on developing fish embryos. Studies involving pregnant mice lacking the Ogg1 enzyme, crucial for repairing DNA damage both in mammals and fish, have shown increased levels of 8-oxo-dG in fetal brains following METH exposure [179]. Offspring subjected to METH exhibit long-term neurodevelopmental deficits, indicating the potential for lasting harm. Given its abundance in zebrafish, it is highly probable that analogous phenomena to those observed in mice may occur in fish populations [180]. It was predicted that the no-effect concentration of the offspring deformations induced by METH calculated from the effective concentration EC10 (24.5 μg/L) and extrapolated Factor (50) was 490 ng/L [137].

It has been proven that exposure to METH profoundly influences the expression of specific genes implicated in the mitogen-activated protein kinases pathway in both fish and mammals. Notably, genes such as mapk14, C-fos, and erk1, alongside Kv2.1, which plays a crucial role in facilitating potassium ion efflux during neuronal apoptosis via p38 mitogen-activated protein kinase-dependent membrane insertion, exhibit noticeable alterations following METH exposure [181]. These effects include increased cleaved-caspase 3 levels and decreased bcl-2/bax ratio [147,182,183]. Furthermore, in addition to its impact on the mitogen-activated protein kinases pathway, METH-induced apoptosis has been linked to the activation of the P53 signaling pathway [89,116]. These pathways serve as critical regulators of cellular survival and death, underscoring the intricate nature of METH’s effects on cellular processes. The involvement of these pathways in METH-induced apoptosis highlights the complexity of the drug’s impact on cellular mechanisms and underscores the importance of further elucidating these processes.

Similarly, exposure to COC and its metabolites (benzoylecgonine and ecgonine methyl) at environmentally relevant concentrations (20 ng/L–1.0 μg/L) has been shown to reduce cell viability, increase DNA fragmentation, activate apoptosis, and alter protein profiles related to cell surface GRP78 in zebrafish and European eel (*Anguilla anguilla*) [82,85,98]. Newer drugs of abuse, like psychoactive phenylamine 25H-NBOH and 25H-NBOMe and metaphedrone, have also been found to bind to the unclassical major groove of ctDNA, leading to conformational changes in the DNA structure and induce genetic damage in zebrafish evaluated by the alkaline comet assay (an increase of tail moment of 48%), with implications for behavior and mobility, e.g., speed (an increase of 49%), total distance moved (an increase of 53%), and absolute turn angle (an increase of 48%) [184,185]. While the effects of metaphedrone were observed at relatively high concentrations (up to 100 µg/L), the findings raise concerns about the potential long-term consequences of exposure to emerging synthetic drugs. Furthermore, research on multidrug users has revealed higher frequencies of nuclear abnormalities in individuals with a history of ID abuse, indicating a correlation between drug consumption and cytogenotoxic damage [177]. 

In addition to neurological and genetic effects, IDs can induce physiological changes in fish organs, such as the liver and kidneys. METH (25 mg/L for 7 days), cocaine. and venlafaxine exposure in zebrafish has been linked to increased oxidative stress and apoptosis in the liver, kidney, and brain tissues of fish [186], the disruption of early brain development, and increased neurogenesis in the hypothalamus, dorsal tuberculum, and preoptic region determined by the higher spatial expression of nrd4 [187], mirroring the hepatotoxic and nephrotoxic effects observed in mammalian models. However, promisingly, interventions such as rosmarinic acid combined with nano-ZnO have shown potential to mitigate the adverse effects of METH [186].

In summary, the documented teratogenic, genotoxic, and cytotoxic effects of IDs on fish populations highlight that drug abuse not only poses significant risks to human health but also has far-reaching consequences for aquatic biodiversity. Elevated frequencies of nuclear abnormalities in fish exposed to IDs, along with altered DNA-methylation and epigenetic effects, emphasize the need for comprehensive approaches to environmental protection.

## 4. Conclusions

The present review provides an overview of the environmental effects of IDs in the context of fish, indicating that they may be exposed to these compounds due to the discharge of wastewater due to insufficient removal during WWTPs processes, leading to ID bioaccumulation and multimechanistic toxicity, affecting fish development and behavior. These ecological effects, which may not be fully recognized by health and environmental authorities, call for multifaceted actions, including education activities, further support for ID use prevention projects, the development of novel wastewater treatment processes to decrease the discharge of trace concentrations of organic compounds, including IDs, and the monitoring of IDs in wastewater and aquatic environments that receive WWP effluents. This is particularly pivotal given that fish play a critical role in maintaining the homeostasis of freshwater and marine ecosystems, their significant dietary role, and the ongoing drug crisis in the post-pandemic reality. 

## Figures and Tables

**Figure 1 pharmaceuticals-17-00537-f001:**
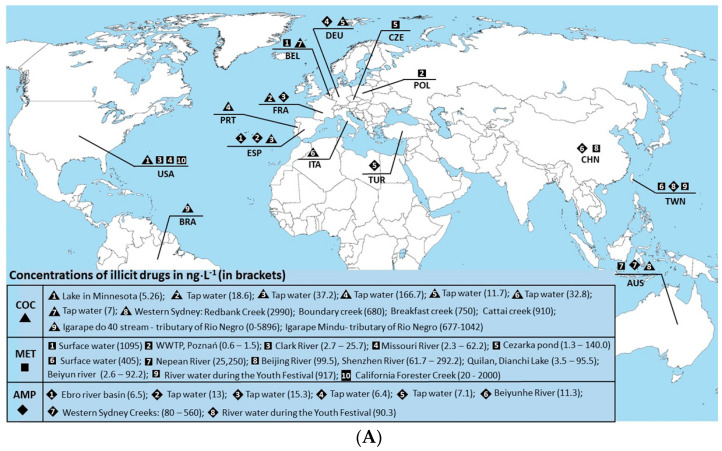
Occurrence and concentration of cocaine, methamphetamine, and amphetamine (**A**) as well as medicine-prescribed drugs (**B**) in rivers and surface water. Three-letter codes represent country names according to ISO 3166-1 [64]. ND means no data. WWTP—wastewater treatment plant.

**Table 2 pharmaceuticals-17-00537-t002:** Occurrence of selected illicit drugs in seawater.

IDs	Enantiomer	Place	Location	Concentration [ng/L]	Ref.
Amphetamine	S(+)-AMP	Scotland	Clyde Estuary (5 different locations)	ND	[76]
	R(−)-AMP			<40	
	R/S(±)-AMP			<30	[77]
		Southern Aegean Sea	Santorini Island INF/EFF	<10.1/<LOD	[78]
		USA	San Francisco Bay	<9.8	[79]
Methamphetamine	S(+)-METH	Scotland	Clyde Estuary (5 different locations)	ND	[76]
	R(−)-METH			ND	
		Southern Aegean Sea	Santorini Island INF/EFF	<LOD	[78]
Cocaine		Brazil Sao Paulo	Santos Bay, surface	12.6–400.5	[80]
			Santos Bay, bottom	29.8–537	
		Southern Aegean Sea	Santorini Island INF/EFF	4.4–49.3/<11.5	[78]
		USA	San Francisco Bay	<2.5	[79]
Benzoylecgonine		Brazil Sao Paulo	Santos Bay, surface	10.9–19.5	[80]
			Santos Bay, bottom	4.6–20.8	
		Southern Aegean Sea	Santorini Island INF/EFF	9.0–104.6/<77.3	[78]
Morphine				1.8–19.8/<2.7	
Methadone				0.9–5.6/<4.5	
9-tetrahydrocannabinol				2.8–31.7/1.2–22.0	
-lysergic acid diethylamide				<1.6	

ND—not detected in at least 50% of all water samples.

## Data Availability

Data sharing is not applicable.

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
