# Peer review of "Illicit Drugs in Surface Waters: How to Get Fish off the Addictive Hook"

_pharmaceuticals, 2024, doi:10.3390/ph17040537_

Round 1

Reviewer 1 Report

Comments and Suggestions for Authors

The manuscript by Prof. Falfushynska, Prof. Rzymsk, and co-authors reviews data on pollution caused by illicit drugs in surface waters and its effect on fish. It is well-written, professionally organized, and elaborates on an understudied issue that deserves more attention. This review will be an ideal point of reference for citations.

I only have some minor issues to be addressed before the publication of this paper in Pharmaceuticals:

1) it would be good to have a list of all abbreviations at the beginning of the manuscript. It's easier to check such a list than search the text to remind yourself what a particular abbreviation means.

2) In L123 the number is reported as mg of BE per 1000 people per day; later mg/100 people/day are reported. Unify the way the units are reported, likely to mg BE/1000 people/day.

3) L126: change to: declining from 39 to 18 mg/1000 people/day (to avoid repetition of the unit)

3) Table 1: change ng L-1 to ng/L

4) Table 1 is not using abbreviation, Table 2 does. Please change abbreviations in Table 2 to full drug names so it is easier to understand, and align the first column to the left.

5) L284: report one decimal number. Check the rest of the manuscript in this regard and correct if necessary.

6) In line with joint recommendations regarding citing authorities describing new taxonomic species (https://riojournal.com/article/94338), please report the author describing the species when reporting a Latin name. For example: Oryzias latipes (Temminck & Schlegel).

Comments on the Quality of English Language

Minor editing of English language required.

Author Response

Thank you for reviewing our work and for positive feedback. We have addressed all minor issues as requested

Reviewer: The manuscript by Prof. Falfushynska, Prof. Rzymsk, and co-authors reviews data on pollution caused by illicit drugs in surface waters and its effect on fish. It is well-written, professionally organized, and elaborates on an understudied issue that deserves more attention. This review will be an ideal point of reference for citations. I only have some minor issues to be addressed before the publication of this paper in Pharmaceuticals:

Authors: Thank you for reviewing our work and for positive feedback. We have addressed all minor issues as requested.

1) it would be good to have a list of all abbreviations at the beginning of the manuscript. It's easier to check such a list than search the text to remind yourself what a particular abbreviation means.

Authors: Added. Moreover, number of abbreviations was reduced.

2) In L123 the number is reported as mg of BE per 1000 people per day; later mg/100 people/day are reported. Unify the way the units are reported, likely to mg BE/1000 people/day.

Authors: Changes here and elsewhere.

3) L126: change to: declining from 39 to 18 mg/1000 people/day (to avoid repetition of the unit)

Authors: Corrected.

3) Table 1: change ng L-1 to ng/L

Authors: Changes here and elsewhere.

4) Table 1 is not using abbreviation, Table 2 does. Please change abbreviations in Table 2 to full drug names so it is easier to understand, and align the first column to the left.

Authors: Done.

5) L284: report one decimal number. Check the rest of the manuscript in this regard and correct if necessary.

6) In line with joint recommendations regarding citing authorities describing new taxonomic species (https://riojournal.com/article/94338), please report the author describing the species when reporting a Latin name. For example: Oryzias latipes (Temminck & Schlegel).

Authors: Done.

Reviewer 2 Report

Comments and Suggestions for Authors

The manuscript “Illicit Drugs in Surface Waters: How to Get Fish Off the Addictive Hook”, authored by Halina Falfushynska * , Piotr Rychter , Anastasiia Boshtova , Yuliia Faidiuk , Nadiia Kasianchuk , Piotr Rzymski submitted to the Special Issue “Zebrafish as a Powerful Tool for Drug Discovery” has adequate and critical content for the scientific community and, consequently, deserve to be published.

Comments and suggestions include:

Page 2, Graphical abstract: give a title to it or include comments on it.

Page 3, line 130: In the paragraph "Comparatively, METH exhibited the highest levels in the USA and Australia, while 130 benzoylecgonine was most prominent in South America", it seems a reference is lacking. I suggest referencing:  Freitas, L.d.A.A.; Radis-Baptista, G. Pharmaceutical Pollution and Disposal of Expired, Unused, and Unwanted Medicines in the Brazilian Context. J. Xenobiot. 2021, 11, 61-76. https://doi.org/10.3390/jox11020005, since Table 2 of the article contains data about cocaine and its metabolites in water bodies.

For convenience to the readers, I suggest including a table listing the kinds of illicit drugs detected, the species of affected fish (or only zebrafish), the major biological (adverse) effects, and the main molecular effect detected, with references in the literature. This table could highlight and summarize the effects of illicit drugs on zebrafish metabolism and behavior.

Author Response

Thank you for reviewing our work and for positive feedback. We have addressed minor issues as requested.

The manuscript “Illicit Drugs in Surface Waters: How to Get Fish Off the Addictive Hook”, authored by Halina Falfushynska * , Piotr Rychter , Anastasiia Boshtova , Yuliia Faidiuk , Nadiia Kasianchuk , Piotr Rzymski submitted to the Special Issue “Zebrafish as a Powerful Tool for Drug Discovery” has adequate and critical content for the scientific community and, consequently, deserve to be published.

Authors: Thank you for reviewing our work and for positive feedback. We have addressed minor issues as requested.

Page 2, Graphical abstract: give a title to it or include comments on it.

Authors: Thank you for your suggestion. The graphical abstract has been reshaped and expanded to include the most common illicit drugs, as well as organismal and cellular disorders they can provoke. In doing so, we have taken into account your comment regarding generalization.

Page 3, line 130: In the paragraph "Comparatively, METH exhibited the highest levels in the USA and Australia, while 130 benzoylecgonine was most prominent in South America", it seems a reference is lacking. I suggest referencing:  Freitas, L.d.A.A.; Radis-Baptista, G. Pharmaceutical Pollution and Disposal of Expired, Unused, and Unwanted Medicines in the Brazilian Context. J. Xenobiot. 2021, 11, 61-76. https://doi.org/10.3390/jox11020005, since Table 2 of the article contains data about cocaine and its metabolites in water bodies.

Authors: Thank you so much for your comment. The information comes from [30], González-Mariño I, Baz-Lomba JA, Alygizakis NA, Andrés-Costa MJ, Bade R, Bannwarth A, Barron LP, Been F, Benaglia L, Berset JD, Bijlsma L, Bodík I, Brenner A, Brock AL, Burgard DA, Castrignanò E, Celma A, Christophoridis CE, Covaci A, Delémont O, de Voogt P, Devault DA, Dias MJ, Emke E, Esseiva P, Fatta-Kassinos D, Fedorova G, Fytianos K, Gerber C, Grabic R, Gracia-Lor E, Grüner S, Gunnar T, Hapeshi E, Heath E, Helm B, Hernández F, Kankaanpaa A, Karolak S, Kasprzyk-Hordern B, Krizman-Matasic I, Lai FY, Lechowicz W, Lopes A, López de Alda M, López-García E, Löve ASC, Mastroianni N, McEneff GL, Montes R, Munro K, Nefau T, Oberacher H, O'Brien JW, Oertel R, Olafsdottir K, Picó Y, Plósz BG, Polesel F, Postigo C, Quintana JB, Ramin P, Reid MJ, Rice J, Rodil R, Salgueiro-González N, Schubert S, Senta I, Simões SM, Sremacki MM, Styszko K, Terzic S, Thomaidis NS, Thomas KV, Tscharke BJ, Udrisard R, van Nuijs ALN, Yargeau V, Zuccato E, Castiglioni S, Ort C. Spatio-temporal assessment of illicit drug use at large scale: evidence from 7 years of international wastewater monitoring. Addiction. 2020 Jan;115(1):109-120. doi: 10.1111/add.14767, had been mentioned one sentence above. The correspondent reference number was added. Also, suggestion you had made can bring additional value for that. That’s why, we added this reference as was recommended.

For convenience to the readers, I suggest including a table listing the kinds of illicit drugs detected, the species of affected fish (or only zebrafish), the major biological (adverse) effects, and the main molecular effect detected, with references in the literature. This table could highlight and summarize the effects of illicit drugs on zebrafish metabolism and behavior.

Authors: Thank you so much for your suggestion. We have modified the graphical abstract as per your recommendation, while choosing to maintain the same number of tables. The graphical abstract highlights the primary adverse effects that may be induced in fish (e.g., oxidative stress, behavior disorders, inflammation), pathways (e.g., TLR, Nrf2-related), and the most common illicit drugs, as suggested by you.